# Bioactivity Profiles of Cytoprotective Short-Chain Quinones

**DOI:** 10.3390/molecules26051382

**Published:** 2021-03-04

**Authors:** Zikai Feng, Monila Nadikudi, Krystel L. Woolley, Ayman L. Hemasa, Sueanne Chear, Jason A. Smith, Nuri Gueven

**Affiliations:** 1School of Pharmacy and Pharmacology, University of Tasmania, Hobart, TAS 7005, Australia; zikai.feng@utas.edu.au (Z.F.); monila.nadikudi@utas.edu.au (M.N.); ayman.hemasa@utas.edu.au (A.L.H.); sueanne.chear@utas.edu.au (S.C.); 2School of Natural Sciences, University of Tasmania, Hobart, TAS 7005, Australia; krystel.woolley@utas.edu.au (K.L.W.); jason.smith@utas.edu.au (J.A.S.)

**Keywords:** mitochondrial dysfunction, short-chain quinone, bioactivity, cytoprotection

## Abstract

Short-chain quinones (SCQs) have been investigated as potential therapeutic candidates against mitochondrial dysfunction, which was largely thought to be associated with the reversible redox characteristics of their active quinone core. We recently reported a library of SCQs, some of which showed potent cytoprotective activity against the mitochondrial complex I inhibitor rotenone in the human hepatocarcinoma cell line HepG2. To better characterize the cytoprotection of SCQs at a molecular level, a bioactivity profile for 103 SCQs with different compound chemistries was generated that included metabolism related markers, redox activity, expression of cytoprotective proteins and oxidative damage. Of all the tested endpoints, a positive correlation with cytoprotection by SCQs in the presence of rotenone was only observed for the NAD(P)H:quinone oxidoreductase 1 (NQO1)-dependent reduction of SCQs, which also correlated with an acute rescue of ATP levels. The results of this study suggest an unexpected mode of action for SCQs that appears to involve a modification of NQO1-dependent signaling rather than a protective effect by the reduced quinone itself. This finding presents a new selection strategy to identify and develop the most promising compounds towards their clinical use.

## 1. Introduction

Mitochondria are essential organelles involved in many cellular processes such as the control of cell death, Ca^2+^-signaling as well as redox- and energy-homeostasis [1,2,3]. Mitochondria provide about 95% of the cellular chemical energy in the form of adenosine triphosphate (ATP) via oxidative phosphorylation (OXPHOS). Any insult or genetic predisposition that impairs mitochondrial function can lead to a range of mitochondrial diseases such as Leber’s hereditary optic neuropathy (LHON), Leigh syndrome (LS) and dominant optic atrophy (DOA) [4]. In addition, mitochondrial dysfunction is also present in a vast number of common inflammatory (i.e., ulcerative colitis) [5], neurodegenerative (i.e., Alzheimer’s disease, Parkinson’s disease, glaucoma, age-related macular degeneration) [6], neuromuscular (i.e., Duchenne muscular dystrophy, multiple sclerosis) [7], and metabolic disorders (i.e., diabetes, obesity) [8], which illustrates that mitochondrial pathology is widespread. However, despite the large number of affected patients, there is an obvious lack of approved drugs that aim to target mitochondrial function directly. This represents a significant unmet medical need and thus, new drug candidates are needed that can be developed into effective and safe medications.

The chemical class of quinones is known for their wide biological activities that are largely due to the reversible redox characteristics of the quinone core [9,10,11]. Naturally occurring quinones such as coenzyme Q_10_ (CoQ_10_) and vitamin K are described as signaling molecules, enzymatic cofactors and antioxidants that are generally well tolerated and are known to be required for normal mitochondrial function [12]. Although this could make these compounds ideal drug candidates to protect normal mitochondrial function, their generally high hydrophobicity is thought to interfere with drug absorption and distribution [13]. Therefore, synthetic quinones such as idebenone, EPI-743 [14], MitoQ [15], SkQ1 [16], KH176 [17] are marketed or are under development. At present, the only synthetic quinone clinically approved for a single mitochondrial disease is the benzoquinone idebenone [18], which showed some activity to protect against vision loss and restored visual acuity and color vision in LHON patients [19,20,21]. Although idebenone is safe and well tolerated in vitro (IC_50_ = 151.7 μM, 24 h) [22] and in vivo (2250 mg/d, 14 d) [23], it is not an ideal drug as it shows low solubility (logP = 1.24, logD = 3.57) and more importantly is characterized by very low metabolic stability in vitro (*t*_1/2_ = 2 h, 40 μM) [24] and in vivo (*t*_1/2_ = 3 h, 150 mg) [25], which significantly restrict its therapeutic potential.

To overcome these significant limitations, we previously described a class of short-chain quinones (SCQs), which centers around a 2,3-disubstituted naphthoquinone core with an amide linkage on the alkyl side chain [26]. Some of these SCQs showed significantly higher cytoprotective activity [26], higher metabolic stability [24] and/or lower cytotoxicity [22] in the human hepatocarcinoma cell line HepG2. This cell line, widely employed in in vitro studies, represents a robust testing platform to offer reproducible outcomes due to its phenotypic stability and unlimited availability [27]. Two SCQs from our library also showed significantly better restoration of visual acuity compared to idebenone in a rat model of diabetic retinopathy [28]. However, in this in vivo model, the two SCQs differed in their molecular activities; while one significantly reduced vascular leakage, the other effectively suppressed oxidative damage [28]. Despite these associations, it is unclear, which bioactivities are causally responsible for the SCQ-dependent cytoprotection observed in vitro and in vivo, which also applies to all other quinones in development. Therefore, the current study approached this problem by characterizing multiple in vitro bioactivities that were previously proposed to be responsible for cytoprotection of quinones. A large number of closely related SCQ compounds with different cytoprotective activity was tested in the same cell line to answer the question whether SCQ-induced cytoprotection can be associated with a single unified mechanism or not. For this purpose, different cellular responses to 103 SCQs **1**–**103** (Table 1), including 24 novel SCQs (**1**–**21**, **33** and **101**–**102**; see Appendix A for details) were assessed with regard to metabolism-related markers, redox activity, expression of cytoprotective proteins, and oxidative damage. These endpoints were correlated to the specific compound chemistries and to their cytoprotective potentials. In contrast to studies that employ only a single molecule, using a large range of closely related compounds allows to correlate varying cytoprotective activities with biological endpoints, which can link specific activities to a chemical family of compounds. This approach can provide essential information towards the underlying mode(s) of action to support the development of selected candidates of this class of compounds towards their clinical use.

## 2. Results

### 2.1. Cytoprotection against Mitochondrial Dysfunction

In the present study, a transient challenge with the mitochondrial complex I inhibitor rotenone reduced HepG2 cellular viability to 26.9 ± 7.9% (Figure 1, dotted line) compared to cells not exposed to rotenone (100%). Under these conditions of rotenone-induced mitochondrial dysfunction, 62 out of the 103 SCQs significantly improved viability (54 compounds, *p* < 0.001; 4 compounds, *p* < 0.01; 4 compounds, *p* < 0.05; Appendix A). Of the 62 cytoprotective compounds, 9 compounds (**35**, **42**, **64**, **68**, **69**, **71**, **92**, **97** and **99**, *p* < 0.001) significantly improved cell viability in the presence of rotenone to >90% compared to cells not treated with rotenone. While more than half of the test compounds protected viability to levels above rotenone-treated cells, 7 compounds (**25**, **30** and **44**, *p* < 0.001; **48** and **51**, *p* < 0.01; **31** and **32**, *p* < 0.05) were cytotoxic and reduced cell viability more than rotenone alone (Appendix A). Of the 7 chemical classes, compounds with an amino alcohol, amino acid, amino ester (*p* < 0.001) or acid (*p* < 0.05) side chains significantly increased viability when compared to rotenone-treated cells. The highest cytoprotection by compounds with an amino ester side chain was significantly higher than those with aliphatic (*p* < 0.001), acid (*p* < 0.05), aliphatic ester (*p* < 0.001) or slight polar (*p* < 0.01) side chains (Figure 1; Appendix A).

### 2.2. Metabolism-Related Markers

#### 2.2.1. Acute Rescue of ATP

One of the dominant parameters affected by mitochondrial dysfunction is cellular ATP synthesis. A mode of action was previously proposed for benzoquinones that relies on the bypass of complex I to restore mitochondrial electron flow in the presence of rotenone, which was termed ATP rescue [10,11,30]. Compared to non-treated HepG2 cells (100%), rotenone rapidly reduced cellular ATP levels to 33.6 ± 11.1% within 1 h (Figure 2a, dotted line). Under these conditions, 54 out of 103 SCQs (52 compounds, *p* < 0.001; 2 compounds, *p* < 0.05; Appendix A) significantly rescued ATP levels to levels above those of rotenone-treated cells (Figure 2a). Of the 54 compounds, 12 compounds (**39**, **40**, **61**, **64**, **67**, **68**, **70**, **71**, **83** and **85**–**87**, *p* < 0.001) significantly rescued ATP levels in the presence of rotenone to >90% compared to cells not treated with rotenone. Of the 7 chemical classes, compounds with amino alcohol, amino ester (*p* < 0.001) or amino acid (*p* < 0.01) side chains significantly increased ATP levels when compared to rotenone-treated cells. The highest rescue of ATP level by compounds with an amino ester side chain was significantly higher than those with aliphatic, acid, amino acid, aliphatic ester or slight polar (*p* < 0.001) side chains (Figure 2a; Appendix A). A correlation was observed between test compound-induced cytoprotection and extent of cellular ATP level rescue (*R*^2^ = 0.44, Figure 2e).

#### 2.2.2. Extracellular Lactate

Under conditions of mitochondrial dysfunction, NAD^+^ is required to maintain the process of glycolysis, which is generated from NADH during the oxidation of pyruvate to lactate [31]. Quinones are reduced to the hydroquinone form in the presence of cellular reductases like NAD(P)H:quinone dehydrogenase 1 (NQO1), a process that also oxidizes NADH to NAD^+^. This quinone-generated NAD^+^ could theoretically be used to maintain glycolysis without the need to generate potentially toxic lactate levels. Alternatively, SCQs could increase glycolysis further to compensate for reduced mitochondrial ATP synthesis, which could increase lactate levels. Therefore, the current study tested if SCQs could affect lactate production of HepG2 cells. In non-treated cells, lactate in cell culture supernatant was measured as 67.3 ± 8.05 µmol/mg protein (100%; Figure 2b). The test compounds altered extracellular lactate levels between 65.1 and 236.6%. Some compounds with aliphatic (**27**, **29** and **33**, *p* < 0.001; **24**, *p* < 0.01; **13**, *p* < 0.05), amino alcohol (**62**, **66** and **76**, *p* < 0.001; **60**, *p* < 0.01; **72**, *p* < 0.05) or slight polar side chains (**51** and **89**, *p* < 0.001; **58**, *p* < 0.05; Appendix A) significantly increased lactate levels compared to the non-treated cells. In contrast, some compounds with aliphatic (**26**, *p* < 0.01), acid (**39**, *p* < 0.01), amino acid (**96**, *p* < 0.01) or slight polar (**103**, *p* < 0.05; Appendix A) side chains significantly decreased lactate levels compared to the untreated cells. However, based on the means of the 7 chemical classes, none significantly altered lactate levels (Figure 2b). No correlation was observed between cytoprotection and extracellular lactate levels (*R*^2^ < 0.1, Figure 2e).

#### 2.2.3. Extracellular β-Hydroxybutyrate

In the absence of carbohydrates, cells also utilize fatty acids to generate ATP via the β-oxidation of lipids that can lead to the accumulation of ketone bodies [32]. In addition, ketone bodies such as β-hydroxybutyrate (BHB) can initiate bioenergetic and mito-hormetic signaling pathways by inhibiting histone deacetylases and by reducing mitochondrial oxidative radical formation [33]. Therefore, the influence SCQs on the accumulation of BHB in the supernatant of HepG2 cell cultures was measured. In non-treated cells, BHB levels in cell culture supernatant were measured as 1.75 ± 0.41 µmol/mg protein (100%; Figure 2c). In comparison, all test compounds except **33** (1132.1%, Figure 2c, outside *y* axis) with an aliphatic side chain altered cellular BHB levels between 60.2 and 311.8%. Some compounds with aliphatic (**23**, **24**, **32** and **33**, *p* < 0.001), amino alcohol (**2**, **14**, **16**, **59**, **60**, **64**, **66**, **69**, **71**–**73**, **75** and **76**, *p* < 0.001; **35** and **67**, *p* < 0.05), acid (**18** and **38**, *p* < 0.001; **39**, *p* < 0.01), amino acid (**11**, **43** and **100**, *p* < 0.001; **93**, *p* < 0.01), amino ester (**81**, **84** and **85**, *p* < 0.001; **12** and **80**, *p* < 0.05) or slight polar (**51**, **55**, **56** and **103**, *p* < 0.001; Appendix A) side chains significantly increased BHB levels when compared to non-treated cells. In contrast, some compounds with aliphatic (**34**, *p* < 0.01), amino alcohol (**1**, *p* < 0.001; **8**, *p* < 0.01) or aliphatic ester (**47**, *p* < 0.05, Appendix A) side chains significantly decreased BHB levels when compared to non-treated cells. Overall, except for compounds with aliphatic or aliphatic ester side chains, all other 5 classes significantly increased BHB levels in the cell culture media (Appendix A). No correlation was observed between cytoprotection and extracellular BHB levels (*R*^2^ < 0.1, Figure 2f).

### 2.3. Redox Activity

It was previously reported that benzoquinones are largely (93.9%) reduced from the quinone to their hydroquinone form by the cytoplasmic enzyme NQO1 [11], while for naphthoquinones, this information was so far not available. It was generally assumed that the hydroquinone form is responsible for cytoprotection, antioxidant function and other beneficial activities. To determine to what extent the reduction of SCQs is NQO1-dependent, its activity in HepG2 cells was inhibited by dicoumarol as previously described [11]. When the reduction of SCQs (Figure 3a, Appendix A) was compared in the absence and presence of dicoumarol, significant differences were observed, with NQO1 dependence between 0.2 and 82.1% (Figure 3b, Appendix A). Of the 7 chemical classes, the highest NQO1 dependence for reduction was seen for compounds with an amino alcohol side chain, which was significantly higher than for those with aliphatic, acid, amino acid or aliphatic ester side chains (*p* < 0.001, Figure 3b, Appendix A). The highest dependence on other reductases was detected for compounds with an aliphatic ester side chain and was significantly higher than those with amino alcohol (*p* < 0.001), amino ester (*p* < 0.01) or slight polar (*p* < 0.05) side chains (Figure 3c, Appendix A). No general correlation between cytoprotection and chemical reduction of test compounds per se was observed (*R*^2^ < 0.1, Figure 3d). However, a mild positive correlation was observed for cytoprotection and NQO1-dependent reduction of quinones. Conversely, cytoprotection negatively correlated with reduction by non-NQO1 reductases (*R*^2^ = 0.23, Figure 3e,f). In addition, a mild positive correlation was also observed for NQO1-dependent reduction and the acute rescue of cellular ATP levels in the presence of rotenone. Similar to cytoprotection, reduction by non-NQO1 reductases negatively correlated with the acute rescue of cellular ATP levels (*R*^2^ = 0.39, Figure 3g,h).

### 2.4. Expression of Cytoprotective Proteins

#### 2.4.1. Cellular Lin28A Expression

Recently, retinal Lin28A protein expression induced by a benzoquinone was reported to be responsible for the cytoprotective activity observed [34]. This finding was extremely surprising, as Lin28A is generally not expressed in adult tissues (except for reproductive tissues), but is known to be involved in differentiation of embryonic tissues. Lin28A overexpression regulates metabolism by enhancing mitochondrial enzyme production, glycolysis and mitochondrial OXPHOS to alleviate mitochondrial dysfunction and to promote tissue repair [35,36,37]. Since this activity could account for the cytoprotective effects observed with our test compounds, their effects on Lin28A levels in HepG2 cells were determined. In our in vitro system, the test compounds only mildly altered Lin28A levels between 94.7 and 107.0% (Figure 4a). Only **74** (7.0 ± 3.9%, *p* < 0.01) with an amino alcohol side chain and **81** (6.0 ± 3.2%, *p* < 0.05) with an amino ester side chain significantly increased Lin28A levels (Appendix A). None of the 7 chemical classes significantly increased Lin28A levels and no statistical significance was observed between the classes (Figure 4a, Appendix A). Since only two significant effects were detected, no general correlation was observed between cytoprotection and Lin28A expression (*R*^2^ < 0.1, Figure 4d).

#### 2.4.2. Cellular Hsp70 Expression

Heat shock protein 70 (Hsp70) is essential for mitochondrial function. It chaperones mitochondrial protein biogenesis, translocates and folds proteins, and prevents their aggregation to maintain mitochondrial proteostasis [38,39,40,41]. Hsp70 also plays critical roles in mitochondrial DNA (mtDNA) maintenance and replication [42] and protects against diabetes, which is associated with mitochondrial dysfunction [43,44]. Since some quinones have been reported to increase Hsp70 levels [45,46,47], the effects of our test compounds on Hsp70 expression in HepG2 cells were assessed. The test compounds altered Hsp70 levels between 92.0 and 167.7% (Figure 4b, Appendix A). Some compounds with aliphatic (**13**, **27** and **30**, *p* < 0.001), amino alcohol (**74**, *p* < 0.01; **66** and **76**, *p* < 0.05), acid (**17**, **37** and **41**, *p* < 0.001; **39**, *p* < 0.05), aliphatic ester (**44**, *p* < 0.05), amino ester (**20**, **79**, **83** and **84**, *p* < 0.001; **21**, *p* < 0.01) or slight polar (**52**, **54**, **58** and **103**, *p* < 0.001; **51**, *p* < 0.05) side chains showed a significant upregulation of Hsp70, while none of the compounds led to a significant downregulation of Hsp70 (Appendix A). Of the 7 chemical classes, only compounds with a slight polar side chain (130.0 ± 23.7%, *p* < 0.001) significantly upregulated Hsp70 expression (Figure 4b, Appendix A). Comparisons between the chemical classes revealed that compounds with a slightly polar side chain also showed significantly higher Hsp70 induction levels compared to those with amino alcohol (*p* < 0.01) or amino acid *(p* < 0.05) side chains (Figure 4b, Appendix A). However, no general correlation between cytoprotection and Hsp70 expression was detected (*R*^2^ < 0.1, Figure 4e).

#### 2.4.3. Tubulin Acetylation

There is significant evidence that inhibition of histone deacetylase 6 (HDAC6) protects against mitochondrial dysfunction by increasing mitochondrial biogenesis [48], stabilizing the cytoskeleton [49], regulating mitochondrial homeostasis [50], increasing oxidative metabolism [51], restoring mitochondrial transport [52], and increasing mitochondrial motility and fusion [53] to sustain cell viability in vitro and in vivo. Previous reports indicated that some short-chain naphthoquinones can inhibit HDAC6 [54]. As HDAC6 inhibition increases tubulin acetylation [55], acetylated tubulin was used as a surrogate marker to detect HDAC6 inhibition in HepG2 cells by our SCQs. The test compounds altered tubulin acetylation between 84.8 and 134.5% (Figure 4c, Appendix A). Some compounds with aliphatic (**23**, **24**, **27** and **34**, *p* < 0.001; **25** and **33**, *p* < 0.01), amino alcohol (**70**, *p* < 0.01; **61**, *p* < 0.05), acid (**37** and **40**, *p* < 0.001), amino acid (**19**, *p* < 0.001), aliphatic ester (**47**, *p* < 0.001; **44**, **45** and **48**, *p* < 0.01; **46**, *p* < 0.05), amino ester (**20** and **83**, *p* < 0.001; **79**, *p* < 0.01; **84** and **85**, *p* < 0.05) or slight polar (**51**, **53**, **54** and **57**, *p* < 0.001) side chains significantly increased tubulin acetylation, while only **74** (86.2 ± 7.8%, *p* < 0.05) with an amino alcohol side chain significantly reduced tubulin acetylation (Appendix A). None of the 7 chemical classes significantly increased tubulin acetylation and no statistical significance was observed between the classes (Figure 4c, Appendix A). No general correlation was observed between cytoprotection and tubulin acetylation levels (*R*^2^ < 0.1, Figure 4f).

### 2.5. Effects on Oxidative Damage

#### 2.5.1. Basal Lipid Peroxidation

The mild correlation between cytoprotection and chemical reduction could suggest that cytoprotection by SCQs against mitochondrial dysfunction involves the redox characteristics of the quinones [10,11]. One mechanistic explanation could be a hormetic form of protection where SCQs induce a sublethal level of damage that induces cytoprotective and mitoprotective pathways. This effect, termed mito-hormesis, can involve the production of oxidative radicals and has been demonstrated in a variety of systems in vitro and in vivo [56,57]. Most SCQs that are developed as potential therapeutics are characterized as antioxidants and consequently have been reported to prevent lipid peroxidation [58,59,60], while some reports revealed that SCQs can act as pro-oxidants at higher concentrations [61,62]. Unlike reported for a range of benzoquinones [10], none of the naphthoquinone test compounds in the current study showed any significant changes of basal lipid peroxidation (BLP) in HepG2 cells when compared to non-treated cells (Figure 5a, Appendix A). Consequently, no correlation was observed between cytoprotection and BLP levels (*R*^2^ < 0.1, Figure 5d).

#### 2.5.2. Oxidative Protein Damage

Given that the redox activity of some quinones can include pro-oxidative behaviour [61,62], we tested our SCQs for their effects on cellular nitrotyrosine levels as a surrogate marker for oxidative protein damage in HepG2 cells. In our system, the test compounds altered oxidized protein levels (100%) between 69.3 and 121.1% (Appendix A). Some compounds with aliphatic (**26** and **101**, *p* < 0.001), amino alcohol (**36** and **63**, *p* < 0.001; **59** and **64**, *p* < 0.01; **1** and **35**, *p* < 0.05), acid (**18** and **39**, *p* < 0.001; **37**, *p* < 0.01), amino acid (**11**, **90**, **96** and **100**, *p* < 0.001; **92**, **93**, **95** and **97**, *p* < 0.01; **94** and **98**, *p* < 0.05), aliphatic ester (**47**, *p* < 0.001; **44**, *p* < 0.01), amino ester (**50**, *p* < 0.01) or slight polar (**57**, *p* < 0.01; **52**, *p* < 0.05) side chains significantly reduced oxidative damage, while only **58** and **103** (*p* < 0.05) with slight polar side chains significantly increased nitrotyrosine levels (Appendix A). Of the 7 chemical classes, only compounds with an amino acid side chain significantly lowered nitrotyrosine levels (79.6 ± 6.2%, *p* < 0.05), while no statistically significant differences were observed between the 7 chemical classes (Figure 5b, Appendix A). No general correlation between cytoprotection and oxidative protein damage was detected (*R*^2^ < 0.1, Figure 5e).

#### 2.5.3. Oxidative DNA Damage

Since the redox activity of some quinones can include pro-oxidative behaviour [61,62], we tested our SCQs for their effects on cellular γ-H2AX levels as a surrogate marker for DNA damage in HepG2 cells. In our system, test compounds mostly altered the number of γ-H_2_AX-positive cells (1.1 ± 0.8% for non-treated cells, Figure 5c, gray dotted line) between 0.2 and 7.3% (Appendix A), except for **103** with a slightly polar side chain (46.0 ± 10.8%, *p* < 0.001). Of the 103 test compounds, only **43** (6.0 ± 1.9%, *p* < 0.05) with an amino acid side chain as well as the epoxide alkylating agents **54** (7.3 ± 3.4%, *p* < 0.01) and **103** with a slightly polar side chain, significantly increased the number of γ-H_2_AX-positive cells (Appendix A). Overall, compounds with a slight polar side chain (9.2 ± 15.0%, *p* < 0.05, Figure 5c, average line and **103** outside *y* axis) significantly elevated the percentage of positive cells on average, while all other chemical classes appeared non-genotoxic (Appendix A). No general correlation between cytoprotection and oxidative DNA damage was detected (*R*^2^ < 0.1, Figure 5f).

### 2.6. Heatmap of Results

All in vitro bioactivities (Figure 1, Figure 2, Figure 3, Figure 4 and Figure 5, Appendix A) and physical properties (Appendix A) of the test compounds **1**–**103** are summarized as a heatmap (Figure 6).

## 3. Discussion

This study aimed to characterize the in vitro bioactivities of a library of 103 recently described short-chain naphthoquinones (SCQs) [22,24,26] to investigate the underlying mechanism(s) of SCQ-dependent cytoprotection. For this purpose, the current study employed not only the previously reported endpoints of cytoprotection and normalization of cellular ATP levels [26], but also assessed additional bioactivities of quinones that are indicative of mitochondrial function, compound bioactivation, expression or activities of cytoprotective proteins (Lin28A, Hsp70, HDAC6), oxidative damage and DNA damage. The results of these endpoint measurements were used to reveal potential correlations with SCQ-induced cytoprotection. While correlations obtained in vitro do not necessarily allow a direct translation to the in vivo situation, the current study aimed to provide a first unbiased insight into the molecular activities of SCQs by utilizing a larger number of test compounds.

A prior study, based on a small numbers of SCQs, suggested that a combination of a naphthoquinone core with selected functional groups attached to a side chain, was responsible for the cytoprotective activity of SCQs [26]. The previous study also concluded that the lipophilic tail moiety is the major determinant of cytoprotection [26], which was confirmed by the current study that observed a large range of cytoprotective activities associated with different side chain chemistries. While most compounds showed some level of cytoprotection against mitochondrial dysfunction, of particular interest were the compounds with an amino ester side chain, all of which significantly protected cellular viability against rotenone with the highest average cytoprotection of all side-chain classes tested. Based on the results of the current study, the structure-activity relationship (SAR) with the most active compounds supports the function of the side chain playing a key role in the activity of the short chain naphthoquinones of this study, which confirms the results of a prior study [26]. The incorporation of an amide into the side chain significantly increased the cytoprotective activity compared to the presence of a polar carboxylic acid or less polar ester linkages. However, at present the exact role of the amide function is not clear. Future studies have to establish if the amide function just changes the polarity of the molecule or if it is directly involved in target binding.

Overall, it is surprising that most endpoints assessed in this study did not correlate with the cytoprotective activity of the test compounds. This suggests that in our test system most endpoints previously attributed to mitoprotection (i.e., extracellular lactate, β-hydroxybutyrate, expression of Lin28A or Hsp70, HDAC6 inhibition, basal lipid peroxidation, oxidative protein or DNA damage) are not responsible for the cytoprotection against rotenone and that other underlying mechanisms are responsible for the cytoprotective effects. While we acknowledge that our results cannot exclude tissue-specific bioactivities, such as neuroprotection, the enzymatic reduction of SCQs and the acute redox-dependent rescue of ATP levels mildly correlated with cytoprotection in the present study. SCQs are believed to be bioactivated by two-electron reductases such as NQO1 to the hydroquinone form upon entering the cell [26]. Despite their reduction in the cytosol, some hydroquinones can donate electrons to the mitochondrial electron transport chain (ETC) to restore proton flux, membrane potential and ATP production under conditions of complex I-deficiency [10,11]. This is achieved by SCQs bypassing the dysfunctional complex I and feeding electrons to complex III of the ETC (Figure 7). Therefore, both cytosolic and mitochondrial activities are required for the acute rescue of ATP levels in vitro. Based on the common quinone core of our test compounds, it was expected that all test compounds are reduced by NQO1 to a similar extent. However, our test compounds exhibited variable levels of reduction by NQO1. This indicates that our SCQs are bioactivated not only by NQO1, but also other reductases that could include vitamin K epoxide reductase (VKOR), as predicted by the shared naphthoquinone moiety with vitamin K [63,64]. Surprisingly, SCQ reduction by NQO1 was positively correlated with cytoprotection, while reduction by other reductases was negatively correlated with SCQ-induced cytoprotection. This was entirely unexpected, as it should not matter how SCQs are bioactivated or where the electrons for their reduction originate from, since only the hydroquinone form was so far thought to be responsible for the protective effects. Our data significantly question this view and suggest a different interpretation that it is not the reduced SCQs themselves that are responsible for the cytoprotective activity but instead their effect on NQO1 and its substrates.

One possible explanation could be that SCQs, by activating NQO1, alter the levels of NQO substrates such as cytoplasmic NADPH, which is subsequently responsible for a cytoprotective activity. For another quinone, dunnione, this mode of action was proposed to be responsible to ameliorate acute pancreatitis on the basis that lower levels of NADPH would result in reduced NADPH oxidase (NOX)-dependent ROS production and reduced tissue damage [65]. Although we did not test cytoplasmic NADPH/NADP^+^ ratios in our cells, it has to be noted that cytoplasmic NADPH is essential for a variety of essential cellular function such as nucleotide synthesis and reactivation of glutathione [66,67]. Therefore, reduced NADPH levels should under physiological conditions lower the concentrations of the most important cellular antioxidant, and consequently increase oxidative stress. In contrast, we did not find any evidence that oxidative damage is associated with the cytoprotective activity of the test compounds. In addition, it is not easily conceivable how reduced cytoplasmic NADPH levels could protect against mitochondrial rotenone toxicity. Therefore, this possibility might apply to certain pathological conditions, while it does not seem to be valid for the test system of this study.

Alternatively, SCQs could affect cellular NADH levels by their interaction with NQO1. Their NQO1-dependent reduction could increase cellular NAD^+^ levels. A previous study observed the activation of the NAD^+^-dependent deacetylase sirtuin 2 (Sirt2) in response to the NQO1 substrate β-lapachone [68]. Although Sirt2 suppresses inflammation [69], stimulate the pentose phosphate pathway [70], it is also associated with neurodegenerative and metabolic diseases and cancer [71]. In this report, Sirt2-dependent deacetylation of microtubules was dependent on NQO1-generated NAD^+^ [68]. Based on our observation that SCQ-induced tubulin acetylation did not correlate with their cytoprotective effects and that overall only minor effects on tubulin acetylation were observed, it is unlikely that the SCQs in the present study affected Sirt2 activity and therefore the NADH/NAD^+^ ratio. It is also unclear how increased NAD^+^ levels, if present at all, could protect cells against rotenone toxicity, without at the same time upregulating lactate levels.

Another explanation why selective reduction of SCQs by NQO1 correlated with their cytoprotective activity could involve a direct modification of NQO1 activities by SCQs. While the current study did not assess the effect of SCQs on NQO1 enzymatic activity, it is important to note that inhibition of NQO1 activity has never been reported as cytoprotective. In contrast, NQO1 inhibition increased the sensitivity of cells to a host of stressors [72] and hence it is unclear how a possible inhibition of NQO1 activity per se, could be responsible to protect against rotenone exposure [73,74]. Therefore, our data suggests that SCQs might either activate NQO1 or provide it with additional functionality (Figure 7). NQO1 is a cytoprotective enzyme that not only detoxifies xenobiotics and displays endogenous superoxide dismutase activity but is also involved in a multitude of cellular signaling events. By acting as proteasomal gatekeeper [75], NQO1 controls protein levels of key signaling molecules such as the tumor suppressor p53 [76], and HIF1 [77]. It is possible that binding of selected SCQs to NQO1 could alter the 3D structure of NQO1, which was described for another NQO1 substrate β-lapachone [68]. The authors of this study proposed that NQO1 acts as a sensor of the cellular redox environment by binding to specific substrates based on an altered protein structure. In this study, NQO1 co-localized with acetylated tubulin during mitosis and it was suggested that this activity would mediate cytoprotection by modifying acetylated tubulin dynamics [68]. Although the present study did not observe major changes to tubulin acetylation, we did not investigate local changes such as on the mitotic spindle. Altered NQO1 structure and activity based upon ligand binding provides a completely novel view on how SCQs could modify cellular signaling by activating NQO1 and controlling its binding to selective proteins to mediate cytoprotection. Similar to the binding of selected proteins, NQO1 also selectively binds to certain mRNAs, a function that for example increases the translation of α1-antitrypsin [78]. Therefore, future studies will investigate if exposure to the cytoprotective SCQs described in this study can mediate the pleiotropic effects of NQO1 by affecting NQO1 binding to proteins or RNAs. We previously reported that the redox activity of the quinone moiety is required but not sufficient for cytoprotection by SCQs [26]. Instead, the cytoprotective activity was localized to the structure of the side chain [26]. Together with the results of the present study, this could suggest that it is the SCQ side chain that generates selectivity for NQO1 to alter its interaction with RNA or protein molecules. This hypothesis will have to be verified in future studies in molecular detail using additional test compounds.

Our results also confirm previous reports that quinone-dependent rescue of ATP levels is dependent on NQO1 [10,11]. Most SCQs that rescued ATP levels were cytoprotective, and consistently, most that left ATP levels unaffected or even decreased them further were not. It is important to note that this does not necessarily imply that rescuing ATP levels is cytoprotective per se. Instead, it is more likely that the rescue of ATP levels is merely a reflection of the ability of quinones to interact with NQO1 and to get reduced. The reduced hydroquinones subsequently bypass complex I by shuttling electrons to complex III of the mitochondrial ETC, which enables ATP production in the presence of rotenone [11].

Collectively, this study attempted a pharmacological approach to characterize chemically closely related SCQs and identified several promising SCQs with interesting in vitro bioactivities. This study aimed to identify a possible mode of action (or lack thereof) by correlating different bioactivities with the corresponding cytoprotective effects for each compound and compound chemical class. The results of the present study question whether SCQ-induced cytoprotection is caused by previously suggested bioactivities such as their antioxidant function. Instead, the results suggest that the mode of action of SCQs could involve a so far unidentified direct modification of NQO1-dependent signaling. The detailed mode of action of SCQs will require confirmation in the presence of rotenone in combination with selected SCQs in future experiments. The current results not only serve as a starting point to elucidate this NQO1-dependent form of cytoprotection, but if confirmed, also enables the future optimization of mitoprotective SCQs. Based on the current status, understanding this mode of action, in combination with detailed in vivo pharmacokinetic and efficacy studies will identify the most mitoprotective, stable and safe candidates that could be developed for a large range of mitochondrial diseases and disorders associated with mitochondrial dysfunction.

## 4. Materials and Methods

### 4.1. Chemicals and Reagents

All SCQ test compounds were synthesized in-house (Chemistry, School of Natural Sciences, University of Tasmania, Hobart, TAS, Australia) with purities > 95% determined by NMR analysis. Dimethyl sulfoxide (DMSO), Dulbecco’s Modified Eagle Medium (DMEM), sodium bicarbonate, glutamic pyruvic transaminase, monopotassium phosphate (KH_2_PO_4_), phenazine methosulphate (PMS), dichlorophenolindophenol (DCPIP), nicotinamide adenine dinucleotide (NAD^+^), tris(hydroxymethyl)aminomethane hydrochloride (Tris-HCl), menadione, celastrol, shikonin, tubastatin, paraformaldehyde (PFA), Tween 20, rat tail collagen, and bovine serum albumin (BSA), rabbit polyclonal anti-Lin28A antibody (SAB2702125), and mouse monoclonal anti-acetyl-tubulin antibody (T7451) were purchased from Sigma-Aldrich (Ryde, NSW, Australia). Fetal bovine serum (FBS), penicillin-streptomycin, ethylenediaminetetraacetic acid (EDTA), trypsin, phosphate buffered saline (PBS), Hanks Balanced Salt Solution (HBSS), BODIPY C11_581/591_, Triton X-100, 4′,6-diamidino-2-phenylindole (DAPI), goat anti-mouse Alexa Fluor 594 secondary antibody (A-11072), and goat anti-mouse Alexa Fluor 488 secondary antibody (A-11029) were obtained from ThermoFisher Scientific (Scoresby, VIC, Australia). D-luciferin and luciferase were obtained from Promega (Alexandria, NSW, Australia). DC Protein Assay Kit was purchased from BioRad Laboratories (Gladesville, NSW, Australia). Water-soluble tetrazolium salt (WST-1) was from Cayman Chemical (Redfern, NSW, Australia). Lactate dehydrogenase was from Cell Sciences (Newburyport, MA, USA). Rabbit monoclonal anti-Hsp70 antibody (EP1007Y) and mouse monoclonal anti-3-nitrotyrosine antibody (ab61392) were from Abcam (Melbourne, VIC, Australia). Mouse monoclonal anti-phospho-Histone H2AX antibody (05-636-I) was from Merck (Kilsyth, VIC, Australia). Cell culture plastics were obtained from Corning (Mulgrave, VIC, Australia), if not stated otherwise.

For all assays, stock solutions of test compounds (SCQs) and reference compounds (10 mM in DMSO) were prepared as single use aliquots and stored at −20 °C until used. All test compounds were used at a final concentration of 10 μM, as previously published [10,79].

### 4.2. Cell Culture

The human hepatocellular carcinoma cell line HepG2 (HB-8065) was obtained from the American Type Culture Collection (ATCC, Manassas, VA, USA). The cells were cultivated in DMEM (10% FBS, 1 g/L glucose, 3.7 g/mL sodium bicarbonate, 100 U/mL penicillin-streptomycin, and 0.584 g/L glutamine) under standard conditions (37 °C, 5% CO_2_, 95% humidity). The cells were routinely passaged twice weekly in T25 cell culture flasks.

### 4.3. Cytoprotection against Mitochondrial Dysfunction

Cytoprotection against mitochondrial dysfunction was measured as previously described [80]. Briefly, 5 × 10^3^ cells/well were preincubated in DMEM, in 96-well plates with test compounds (10 μM in DMEM) for 2 days prior to being challenged by the mitochondrial complex I inhibitor rotenone (1 μM in HBSS, 7 h). After post-incubation with only test compounds (10 μM in HBSS) for an additional 24 h, cell viability was quantified by analyzing ATP content per well using a luciferase-based reaction as previously described [80]. Data was standardized on the non-treated control (no rotenone) and expressed as mean ± standard deviation (SD) of 6 replicates from 3 independent experiments.

### 4.4. Acute Rescue of ATP Levels

Acute rescue of ATP levels in the presence of rotenone was measured as previously described [10]. Briefly, 1.5 × 10^4^ cells/well were seeded in 96-well plates in DMEM. After 24 h, cells were simultaneously incubated with test compounds (10 μM) and rotenone (10 μM) or rotenone alone for 1 h in glucose-free DMEM before ATP levels were measured using a luciferase-based reaction as previously described [10]. Data was standardized on the non-treated control (no rotenone) and expressed as mean ± standard deviation (SD) of 6 replicates from 3 independent experiments.

### 4.5. Extracellular Lactate

To quantify the effects of the test compounds on extracellular lactate levels, cells were seeded at a density of 1 × 10^5^ cells/well in 6-well plates in DMEM. After 24 h, media was replaced with challenge media (DMEM supplemented with 25 mM glucose, 8 mM l-glutamine, 100 U/mL penicillin-streptomycin and 1 mM pyruvate). After 48 h, cell culture media was collected for lactate measurement and the remaining cells were lysed at room temperature (RT) in lysis buffer (0.5% Triton X-100/PBS) for protein measurement using the DC Protein Assay as recommended by the manufacturer. Lactate was measured as previously described [11]. Briefly, 10 μL of collected cell culture media was added to 90 μL of reaction buffer (10 mM KH_2_PO_4_, pH 7.8, 1 mg/mL BSA, 0.5 mM PMS, 2 mM EDTA, 0.3 mM DCPIP, 0.08 mM NAD^+^, 0.5 U/mL glutamic pyruvic transaminase, 1.5 mM glutamate, 1.25 U/mL lactate dehydrogenase) in transparent 96-well plates including lactate standards from 3 to 23 mM. Absorbance at 600 nm was measured using a plate reader (Multiskan Go, ThermoFisher Scientific, Scoresby, VIC, Australia) every 2 min over a period of 100 min at 30 °C. Data was standardized on protein levels and was expressed as percentage Lactate/Protein compared to non-treated cells. Data represents the mean ± SD of 6 replicates from 3 independent experiments.

### 4.6. Extracellular β-Hydroxybutyrate

Effect of test compounds on β-hydroxybutyrate (BHB) in the cell culture supernatant was assessed by seeding 1 × 10^4^ cells/well in 96-well plates in DMEM. After 24 h, media was replaced with DMEM containing test compounds. After 72 h, supernatant was collected. To 50 μL of reaction mixture (containing 0.5 mM PMS, 2.5 mM NAD^+^, 0.00625 U BHB-dehydrogenase, 10 μM DCPIP in 100 mM Tris-HCl buffer), 50 μL of cell supernatant was added to initiate the reaction. Absorbance at 600 nm was measured every 30 s for 10 min at RT using a plate reader (Multiskan Go., ThermoFisher Scientific, Scoresby, VIC, Australia). A BHB standard curve with BHB concentrations from 0.15 to 5 mM was used to calculate BHB levels in the supernatant of test compounds-treated cells. Data represents the mean ± SD of 6 replicates from 3 independent experiments.

### 4.7. Reduction of Test Compounds

To measure cellular reduction of test compounds in vitro, 1 × 10^4^ cells/well were seeded in 96-well plates in DMEM. After 6 h, media was replaced with DMEM containing 0.3 g/L glucose and 2% FBS and incubated for 18 h. Subsequently, media was replaced with DMEM with/without test compounds with/without dicoumarol (10 μM) and incubated for 1 h. Finally, media was replaced with HBSS with/without test compounds containing a water-soluble, cell-impermeable redox tetrazolium dye (WST-1, 450 μM) and absorbance at 450 nm was measured every 2 min for 2 h at 37 °C using a plate reader (Multiskan Go, ThermoFisher Scientific, Scoresby, VIC, Australia) as previously described [81,82]. Total reduction of test compounds = maximum absorbance in the absence of dicoumarol. Reduction by NQO1 (%) = maximum absorbance in the presence of dicoumarol divided by that in the absence of dicoumarol, × 100%. Reduction by other reductases (%) = 100% − Reduction by NQO1 (%). Data represents the mean ± SD of 6 replicates from 3 independent experiments.

### 4.8. Cellular Lin28A Expression

To measure cellular Lin28A protein levels, 5 × 10^3^ cells/well were seeded in 100 µL serum-free DMEM in 384-well plates (781091, µClear, Greiner, Ryde, NSW, Australia) pre-coated with rat tail collagen (1:20 in HBSS, pH 7.4, 50 µL, 45 min) and left to adhere overnight. Subsequently, cells were treated with test compounds (in HBSS, 100 μL) for 24 h. After fixation (4% PFA/PBS, 50 µL, 10 min), permeabilization (0.5% Triton X-100/PBS, 50 µL, 10 min) and blocking (5% FBS + 5% BSA in PBS, 50 µL, 1 h), cells were exposed to rabbit polyclonal anti-Lin28A antibody (1:500 in blocking buffer, overnight). After washing with PBST (0.1% Tween-20/PBS, 50 μL, 5 min, 3×), cells were exposed to goat anti-rabbit Alexa Fluor 594 secondary antibody (1:10,000 in PBST, 15 µL, 1 h) in the dark. After another 3× washing with PBST, nuclei were counter-stained with DAPI (1:10,000 in PBST, 15 µL, 2 min; Appendix A). After 3× washing with PBST, cells were stored in 50 μL PBS for high content imaging using an INCell 2200 analyzer (10× magnification, GE Healthcare, Rydalmere, NSW, Australia). Average cellular Lin28A intensity was automatically quantified on each acquired image using IN Carta image analysis software (GE Healthcare, Rydalmere, NSW, Australia). Data was standardized on the non-treated control (100%) and expressed as mean ± SD of at least quadruplicates from one assay. At least 1 × 10^3^ cells were analyzed separately per treatment.

### 4.9. Cellular Hsp70 Expression

To measure cellular Hsp70 protein levels, cells were seeded, treated, fixed, and permeabilized as described under 4.6. Celastrol [83] and shikonin [84] were used as positive control compounds. After blocking, cells were exposed to rabbit monoclonal anti-Hsp70 antibody (1:1000 in blocking buffer, 15 µL, overnight). After exposure to goat anti-rabbit Alexa Fluor 594 secondary antibody (1:10,000, 15 µL, 1 h), cells were stained using DAPI and stored in PBS (Appendix A), images were aquired using an INCell 2200 analyzer (10× magnification) and analyzed using IN Carta image analysis software as described above (GE Healthcare, Rydalmere, NSW, Australia). Average cellular Hsp70 intensity was automatically quantified for each acquired image. Data were standardized on the non-treated control (100%) and expressed as mean ± SD of at least quadruplicates from one assay. At least 1 × 10^3^ cells were analyzed for each treatment.

### 4.10. Quantification of Acetylated Tubulin

To measure cellular acetyl-tubulin levels, cells were seeded, treated, fixed, and permeabilized as described above. Tubastatin was used as a positive control [85]. After blocking, cells were exposed to mouse monoclonal anti-acetyl-tubulin antibody (1:1000 in blocking buffer, 15 µL, overnight). After exposure to goat anti-mouse Alexa Fluor 488 secondary antibody (1:10,000, 15 µL, 1 h), cells were stained using DAPI and stored in PBS (Appendix A), images were aquired using an INCell 2200 analyzer (10× magnification) and analyzed using IN Carta image analysis software as described above (GE Healthcare, Rydalmere, NSW, Australia). Average cellular acetyl-tubulin intensity was automatically quantified for each acquired images. Data were standardized over the non-treated control (100%) and expressed as mean ± SD of at least quadruplicates from one assay. At least 1 × 10^3^ cells were analyzed for each treatment.

### 4.11. Basal Lipid Peroxidation

To assess the effects of the test compounds on basal levels of lipid peroxidation, 2 × 10^4^ cells/well were seeded in black 96-well plates in DMEM. After 24 h, the media was removed, and cells were loaded with 10 µM dye solution (1% BODIPY C11_581/591_ in 100 µL HBSS per well) for 30 min. Subsequently, the dye solution was replaced with 100 µL HBSS with/without test compounds and incubated for 1 h. After the cells were washed 3× with PBS, fluorescence was measured (Ex/Em 490/520 and Ex/Em 490/600 in 50 µL PBS) using a plate reader (Fluoroskan Ascent, ThermoFisher Scientific, Scoresby, VIC, Australia) [10]. Fluorescence ratios were calculated and presented as percentage of non-treated control cells. Data represents the mean ± SD of 6 replicates from 3 independent experiments.

### 4.12. Quantification of Oxidative Protein Damage

To assess if the test compounds induce oxidative damage, cells were seeded, treated, fixed, and permeabilized as described above. Shikonin was used as a positive control [86]. After blocking, cells were exposed to mouse monoclonal anti-3-nitrotyrosine antibody (1:500 in blocking buffer, 15 µL, overnight). After exposure to goat anti-mouse Alexa Fluor 488 secondary antibody (1:10,000, 15 µL, 1 h), cells were stained using DAPI and stored in PBS (Appendix A), images were aquired using an INCell 2200 analyzer (10× magnification) and analyzed using IN Carta image analysis software as described above (GE Healthcare, Rydalmere, NSW, Australia). Average cellular nitrotyrosine intensity was automatically quantified for each acquired images. Data were standardized over the non-treated control (100%) and expressed as mean ± SD of at least 8 replicates from one assay. At least 2 × 10^3^ cells were analyzed for each treatment.

### 4.13. Quantification of Oxidative DNA Damage

To assess if the test compounds induce some level of genotoxicity, cells were seeded as previously described and treated with 10 μM test compounds. Since γ-H2AX signal can decrease over time due to DNA repair, this assay captured SCQ-induced DNA damage after a short treatment period of 4 h. Celastrol was used as a positive control [87]. After fixation, permeabilization and blocking, cells were exposed to mouse monoclonal anti-phospho-Histone H_2_AX antibody (1:1000 in blocking buffer, 15 µL, overnight). After exposure to goat anti-mouse Alexa Fluor 488 secondary antibody (1:10,000, 15 µL, 1 h), cells were stained using DAPI and stored in PBS (Appendix A), images were aquired using an INCell 2200 analyzer (10× magnification) and analyzed using IN Carta image analysis software as described above (GE Healthcare, Rydalmere, NSW, Australia). The number of γ-H_2_AX-positive cells was automatically quantified for all acquired images. Percentage γ-H_2_AX-positive cells was expressed as mean ± SD of at least quadruplicates from one assay. At least 500 cells were analyzed for each treatment.

### 4.14. Statistical Analysis

One- or two-way ANOVA followed by Dunnett’s multiple comparison post-test was performed using GraphPad Prism (version 8.2.1, San Diego, CA, USA) to compare test compounds and control(s) or between chemical classes: *** *p* < 0.001, ** *p* < 0.01, * *p* < 0.05, otherwise non-significant (Appendix A).

## Figures and Tables

**Figure 1 molecules-26-01382-f001:**
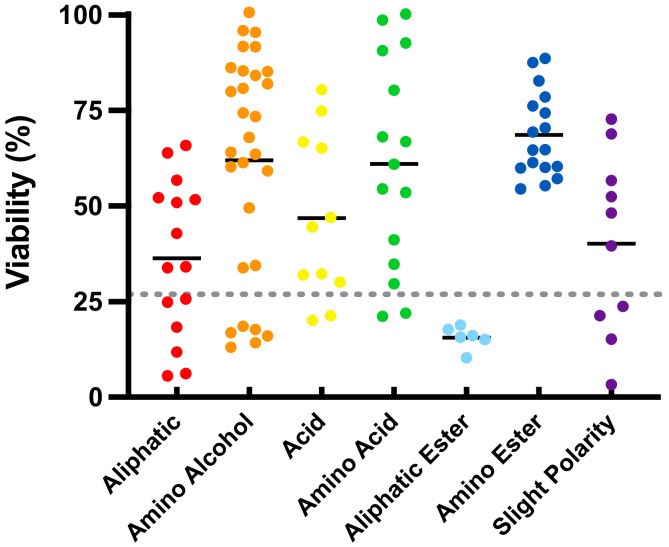
Cytoprotection of short-chain quinones (SCQs) belonging to different chemical classes against mitochondrial dysfunction in HepG2 cells. Each point represents the average responses from several independent experiments for one SCQ. Black solid lines represent the mean for each chemical class. Error bars were omitted for clarity (for detailed information see Appendix A). Gray dotted line represents the effect of rotenone.

**Figure 2 molecules-26-01382-f002:**
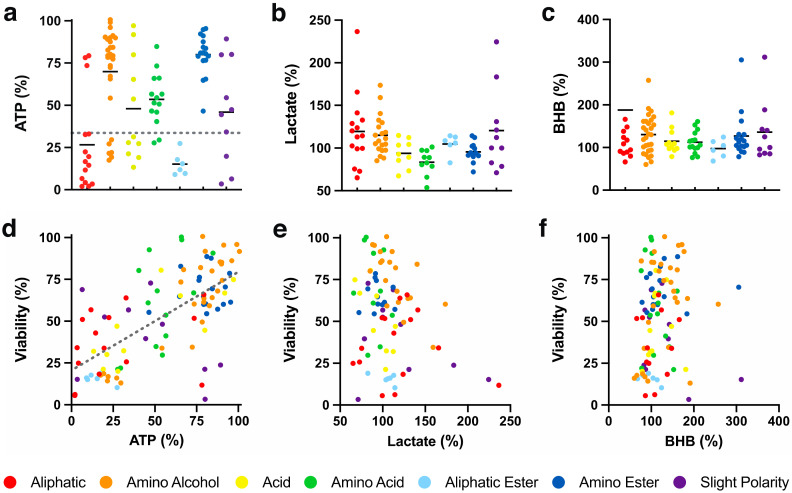
Effect of SCQs on metabolism-related markers in HepG2 cells. (**a**) Cellular ATP levels (gray dotted line represents the effect of rotenone), (**b**) extracellular lactate levels of SCQs, (**c**) extracellular β-hydroxybutyrate (BHB), and (**d**–**f**) their correlations with SCQ-protected viability. Each point represents the average responses from independent experiments for one SCQ. Error bars were omitted for clarity (for detailed information see Appendix A). Linear regression was generated using GraphPad Prism (version 8.2.1, San Diego, CA, USA).

**Figure 3 molecules-26-01382-f003:**
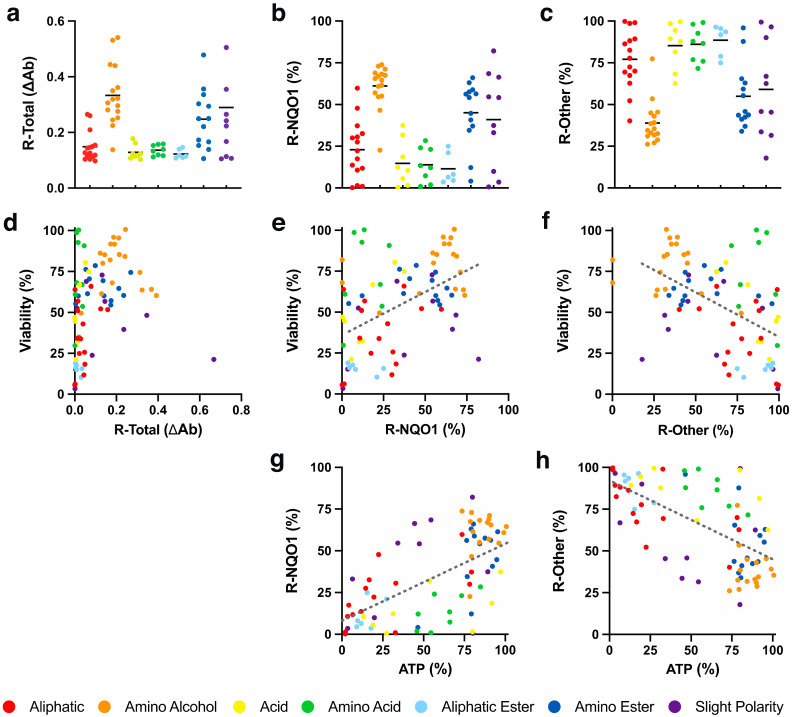
Redox activity of SCQs in HepG2 cells. (**a**) Total reduction of quinones (R-Total), (**b**) reduction by NQO1 (R-NQO1), (**c**) reduction by other reductases (R-Other) and their correlations with (**d**–**f**) SCQ-protected viability and (**g**,**h**) acute rescue of ATP levels. Each dot represents the average responses from independent experiments for one SCQ. Error bars were omitted for reasons of clarity (for detailed information see Appendix A). Linear regressions were generated using GraphPad Prism (version 8.2.1, San Diego, CA, USA).

**Figure 4 molecules-26-01382-f004:**
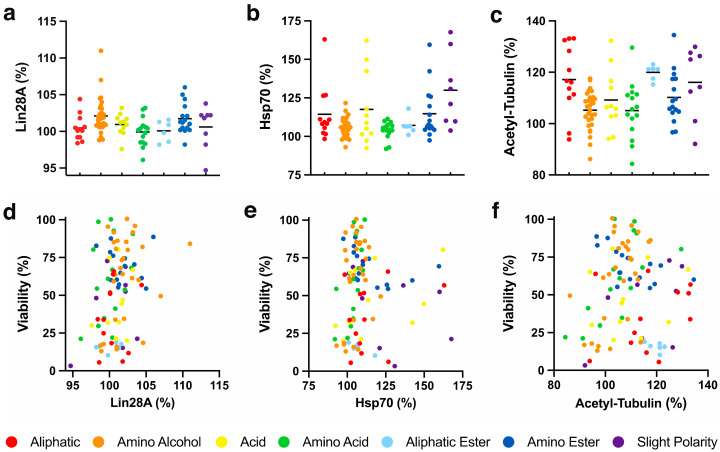
SCQ-induced expression of cytoprotective proteins in HepG2 cells. SCQ-induced (**a**) Lin28A levels; (**b**) Hsp70 levels; (**c**) acetylated tubulin (acetyl-tubulin) levels, and their (**d**–**f**) correlations with SCQ-protected viability. Each point represents the average responses from several independent experiments for one SCQ. Black solid lines represent the mean for each chemical class. Error bars were omitted for clarity (for detailed information see Appendix A).

**Figure 5 molecules-26-01382-f005:**
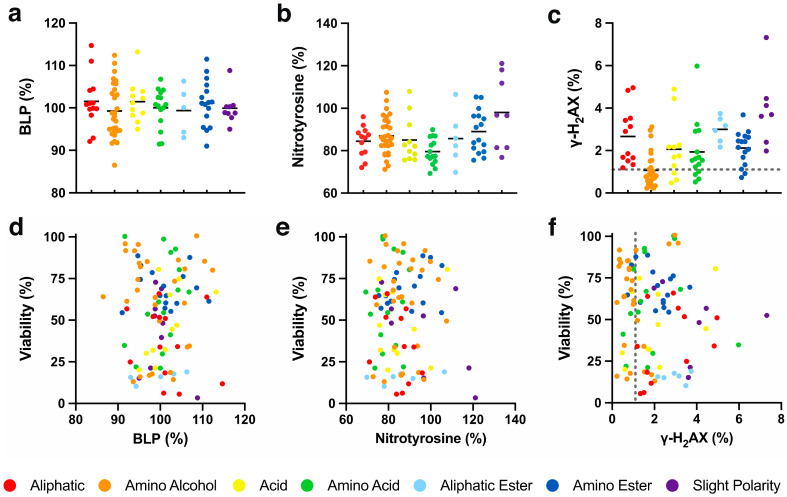
Effect of SCQs on oxidative damage in HepG2 cells. (**a**) basal lipid peroxidation, (**b**) nitrotyrosine levels, (**c**) γ-H_2_AX-positive cells, and their (**d**–**f**) correlations with SCQ-protected viability. Each point represents the average responses from several independent experiments for one SCQ. Black solid lines represent the mean for each chemical class. Error bars were omitted for clarity (for detailed information see Appendix A). Gray dotted lines represent effect on non-treated cells.

**Figure 6 molecules-26-01382-f006:**
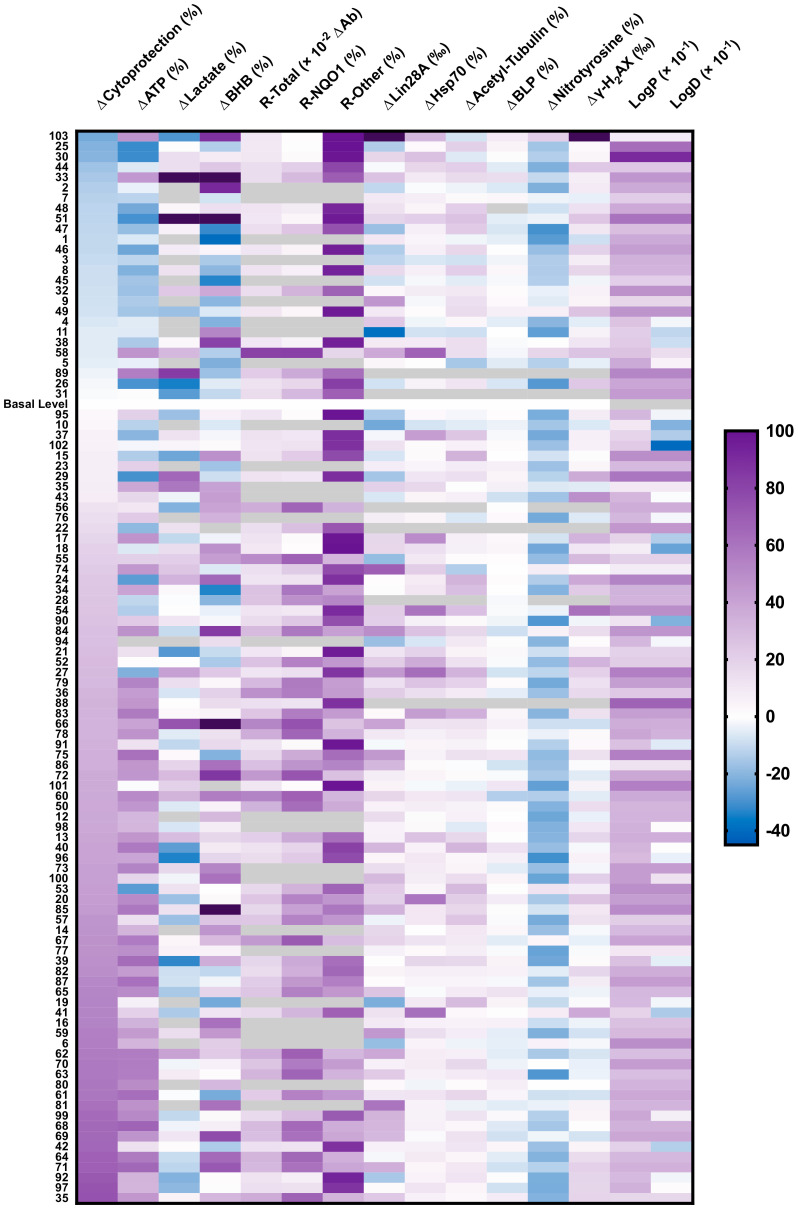
Heatmap of results shown in Figure 1, Figure 2, Figure 3, Figure 4 and Figure 5 and Appendix A. Each column represents one parameter and each line represents one short-chain quinone (SCQ). For some compounds, not all parameters could be assessed (gray boxes). Data expressed as the mean of multiple independent experiments. ΔCytoprotection (%) = SCQ-protected viability (%)—basal viability level (rotenone-treated, 26.9%); ΔATP (%) = SCQ-protected ATP level (%)—basal ATP level (rotenone-treated, 33.6%); logP, partition coefficient; logD, distribution coefficient; BHB, β-hydroxybutyrate; BLP, basal lipid peroxidation; R-Total, total reduction of quinone; R-NQO1, reduction of quinone by NQO1; R-Other, reduction by other reductases; γ-H2AX, γ-H_2_AX-positive cells. Raw data available in Appendix A.

**Figure 7 molecules-26-01382-f007:**
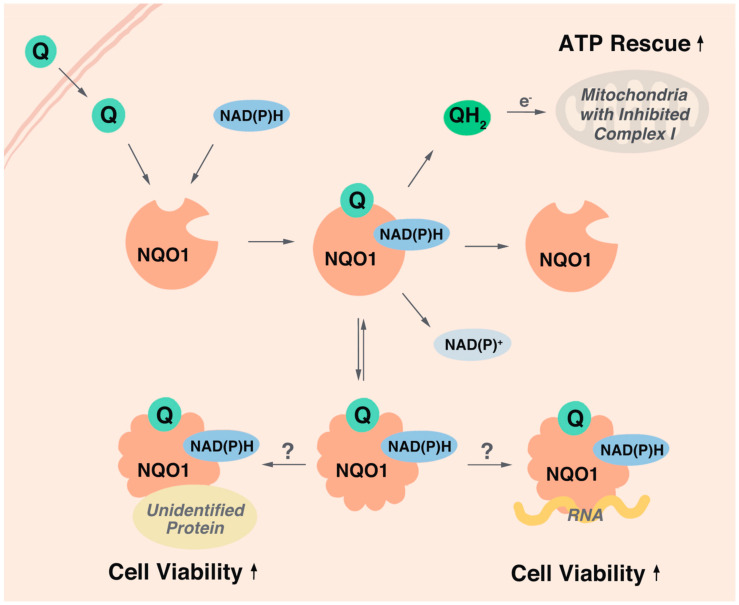
Schematic representation of ligand-induced structural change of NQO1 and hypothetical mechanism for SCQ-induced cytoprotection. Q, short-chain quinone; NQO1, NAD(P)H:quinone oxidoreductase 1; QH_2_, short-chain hydroquinone.

**Table 1 molecules-26-01382-t001:** Short-chain quinone (SCQ) test compounds.

Structure	#	*n*	R	References	#	*n*	R	References	#	*n*	R	References
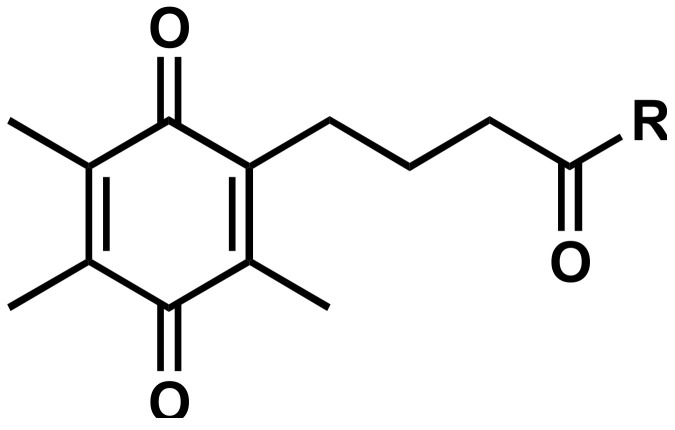	**1**	-	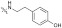	-	**2**	-	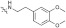	-	**3**	-	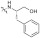	-
**4**	-		-	**5**	-	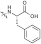	-	**6**	-	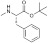	-
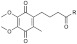	**7**	-	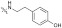	-	**8**	-	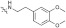	-	**9**	-	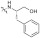	-
**10**	-		-	**11**	-	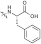	-	**12**	-	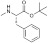	-
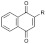	**13**	-	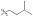	-	**14**	-	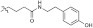	-	**15**	-	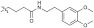	-
**16**	-	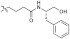	-	**17**	-	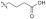	-	**18**	-	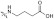	-
**19**	-	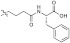	-	**20**	-	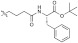	-	**21**	-	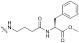	-
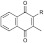	**22**	-	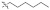	[26]	**23**	-		[26]	**24**	-	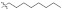	[26]
**25**	-		[26]	**26**	-	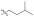	[26]	**27**	-	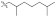	[26]
**28**	-		[26]	**29**	-	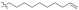	[26]	**30**	-	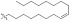	[26]
**31**	-	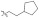	[26]	**32**	-	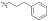	[26]	**33**	-	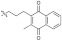	-
**34**	-	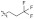	[26]	**35**	-	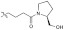	[22,24,26]	**36**	-	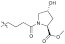	[26]
**37**	-	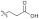	[26]	**38**	-	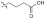	[26]	**39**	-	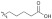	[26]
**40**	-	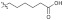	[26]	**41**	-	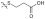	[24,26]	**42**	-	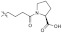	[22,24,26]
**43**	-	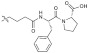	[26]	**44**	-	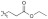	[26]	**45**	-	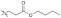	[26]
**46**	-	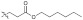	[26]	**47**	-	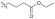	[26]	**48**	-	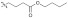	[26]
**49**	-	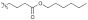	[26]	**50**	-	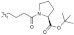	[26]	**51**	-	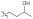	[26]
**52**	-	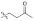	[26]	**53**	-	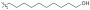	[26]	**54**	-	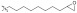	[26]
**55**	-	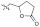	[26]	**56**	-	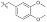	[26]	**57**	-	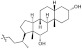	[26]
**58**	-	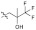	[26]								
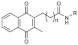	**59**	2		[24,26]	**60**	2	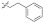	[26]	**61**	2	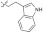	[26]
**62**	2	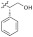	[22,24,26]	**63**	2	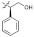	[22,24,26]	**64**	2	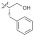	[22,24,26]
**65**	2	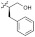	[24,26]	**66**	3	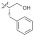	[26]	**67**	4	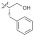	[26]
**68**	2	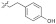	[22,24,26]	**69**	3	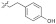	[22,24,26]	**70**	4	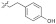	[24,26]
**71**	2	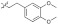	[22,24,26,28]	**72**	3	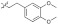	[26]	**73**	4	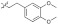	[26]
**74**	2	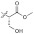	[26]	**75**	2	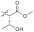	[26]	**76**	2	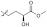	[26]
**77**	2	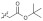	[26]	**78**	2	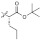	[26]	**79**	2	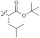	[26]
**80**	2	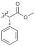	[24,26]	**81**	2	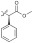	[26]	**82**	2	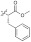	[26]
**83**	1	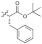	[26]	**84**	2	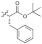	[26]	**85**	3	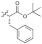	[26]
**86**	4	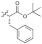	[26]	**87**	2	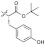	[26]	**88**	2	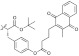	[26]
**89**	2	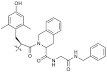	[26]								
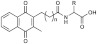	**90**	2	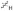	[26]	**91**	2		[26]	**92**	2	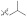	[22,24,26]
**93**	2	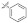	[26]	**94**	2	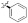	[26]	**95**	2	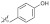	[26]
**96**	1	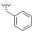	[26]	**97**	2	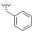	[22,24,26,28]	**98**	2	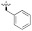	[26]
**99**	3	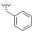	[22,24,26]	**100**	4	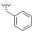	[26]				
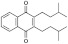	**101**	-	-	-								
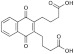	**102**	-	-	-								
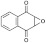	**103**	-	-	[29]								

SCQs are characterized into seven chemical classes: aliphatic, amino alcohol, acid, amino acid, aliphatic ester, amino ester, and slight polarity (see Appendix A for details).

## Data Availability

Data is available in Appendix A.

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
