# Peer review of "Bioactivity Profiles of Cytoprotective Short-Chain Quinones"

_molecules, 2021, doi:10.3390/molecules26051382_

Round 1
Reviewer 1 Report
The manuscript by Feng et al reports on “bioactivity profiles of cytoprotective short-chain quinones (SCQs)”. The authors extensively investigated the ability of SCQs to protect cells at a molecular level using a series of well-described and performed experiments. Overall, the study suggested an unexpected mode of action for SCQs pertaining to the modification of NQO1-dependent signaling. This finding presents potentially a new strategy to identify and develop clinically relevant compounds. I recommend accepting the manuscript as it is.
Author Response
We sincerely appreciate the reviewer for assessing and accepting our manuscript.
Reviewer 2 Report
The authors presented the results of a large-scale study of the cytoprotective activity of more than 100 substances. The authors also made an attempt to discover the relationship between the cytoprotective effect of substances and their effect on some targets. However, this attempt was almost unsuccessful. The authors tried to establish the relationship between cytoprotective activity and action on some cellular targets not for specific active molecules, but for groups of substances with a similar chemical structure. In my opinion, this tactic was doomed to failure in advance. I believe that detailed studies of the mechanism of action are advisable to undertake only for substances that have shown any cytoprotective effect. In each group of the studied substances, both active and inactive ones were identified, therefore, attempts to establish a relationship between the cytoprotective effect of a group of substances as a whole and the effect on some aspects of cellular life will be ineffective.
The authors investigated several aspects of the cytoprotective action of substances (a number of proteins, lipid peroxidation and others). However, such an aspect of the toxic effect of rotenone as an increase in the level of reactive oxygen species in cells remained outside the attention of the authors. Nevertheless, a very high ROS level in the first hours of rotenone action (up to 200 percent compared to the control in the first hour) is one of the main damaging factors. If the substances studied by the authors neutralize this level of ROS as direct antioxidants or by activating the enzymatic antioxidant defense of the cell, then this determines their cytoprotective effect. Mitochondrial dysfunction, which the authors write about in the manuscript, also requires more attention from the authors, in my opinion. In the Introduction section, the authors write about the role of mitochondrial dysfunction in the development of pathological processes and the importance of drugs aimed at compensating for mitochondrial dysfunction. However, in their experiments, the authors did not assess the effect of substances directly on mitochondria, at least on the mitochondrial membrane potential.
Thus, I believe that the manuscript needs improvement. Authors should to bring the Introduction and those experiments, the results of which they represent, into logical correspondence.
I also believe that conclusions about the most active molecules among the studied ones should be made, since in the future their chemical structures will attract interest for citation, and not the relationship between the activities of a group of compounds.
Authors should add information about the cell line used in the work in all sections of the manuscript: Abstract, Introduction and Results, as well as in the captions of Figures and Tables. Moreover, it is necessary to argue in Introduction (or first part of Results) section why hepatocellular carcinoma cell line was chosen for the investigation.
Paragraphs 4.7, 4.8, 4.10 (Methods section) lack the necessary details regarding the equipments and software used, as well as the reagent manufacturer.
Author Response
We thank the reviewer for reviewing our manuscript and providing us with thoughtful comments. Please refer to the point-by-point responses in the document attached.

Reviewer 3 Report
The researchers created the SCQ library which has already been described in their previous work (synthesis, toxicity, acute rescue of ATP levels and cytoprotective activity). In this paper, the authors attempted to investigate the mechanism of action of short-chain quinones (SCQs) as potential therapeutic candidates against mitochondrial dysfunction. The current work was supplemented, i.a., by research on metabolism-related markers, redox activity, expression of cytoprotective proteins and oxidative damage. Ultimately, it was not possible to clearly define the mechanism of action of the tested compounds, this suggests that in the test system most endpoints previously attributed to mitoprotection (i.e. extracellular lactate, β-hydroxybutyrate, expression of Lin28A or Hsp70, HDAC6 inhibition, basal lipid peroxidation, oxidative protein or DNA damage) are not responsible for the cytoprotection against rotenone and that other underlying mechanism are responsible for the cytoprotective effects.
The authors used adequate research methods. The manuscript was prepared carefully. The authors cite the most important literature. The results are well presented and supported by the data.
My most serious objection against this manuscript is that in Table 1, most of the compounds have references to the literature which suggests that the compounds have already been described in earlier works. There are also chemical compounds without any reference (i.e. 1-21, 33, 101-103) that would suggest that the compounds are new, but no data is available on this in Section 4. Materials and Methods or Supplementary Materials. This should necessarily be explained and supplemented If compounds have not been described anywhere, the physicochemical analysis should be added, if they have been described, references should be added.
On lines 76-77, I think this is the wrong letter of the figure.
In line 131 should be amino ester, not "animo" ester.
Author Response
We appreciate the thorough review of our manuscript and agree with the reviewer. Please refer to the point-by-point responses in the document attached.

Round 2
Reviewer 2 Report
The authors did answer some particular questions and made a number of changes to the manuscript. However, in principle, the manuscript remained incomprehensible to me. The authors investigated the cytoprotective properties of substances against rotenone and measured the rotenone-treated cell viability. But have all the other experiments been conducted without rotenone? If so, then the authors' attempts to establish a correlation between cytoprotection toward rotenone-treated cells and the effect on individual cellular targets in rotenone-nontreated cells are doubtful and even erroneous.
If we do not take into account the attempts to establish this correlation, the study of the bioactivity profile of a large number of structurally related substances is quite interesting if conclusions are drawn about the effect of chemical structure on their activity.
Moreover, the concentration of rotenone should be added in the text.
Thus, I believe that the logic of the manuscript aimed at establishing a correlation between cytoprotection and action on individual targets in untreated cells is wrong. If the authors rewrite the manuscript and simply present the results of studying the various types of activity of substances, it will look more understandable and suitable for publication.
Author Response
We sincerely acknowledge and appreciate the reviewer for processing our revisions. We are very happy to include the rotenone concentration (as suggested by the referee) as an essential part of the manuscript and we apologize for the omission. However, to exclude the viability data as suggested, would imply a major revision with changes to all parts of the manuscript, including all correlation figures, as well as a complete re-write of introduction and discussion. We believe that the informed reader will come to the same conclusion as the referee (that "the manuscript is quite interesting if conclusions are drawn about the effect of chemical structure on their activity" which we provided) without removing the correlations. It is surprising to us that this is the first reviewer that raised this point. This also includes one of our prior manuscripts that successfully used this approach for a related class of compounds, which has been well cited. Therefore, we believe that the reviewer may not like to have largely negative data published, which is a bias that is recognized as a major drawback to scientific progress. In this particular case however, our data does show a correlation with the NQO1-dependent reduction of the test compounds, so we cannot follow the reviewer’s argument that our approach was "doomed to fail". Therefore, we contacted the Academic Editor to see if it would be possible to receive some guidance with regards to which minor revisions the Academic Editor would like to see to be made to the manuscript. Subsequently, we received a response from the Academic Editor that “it is enough that the authors add the concentration of rotenone”. For this purpose, we have included two new text sections into the Materials and Methods (i.e. 4.3. Cytoprotection against Mitochondrial Dysfunction and 4.4. Acute Rescue of ATP Levels) that point out the concentrations of rotenone (lines 427-443).This manuscript is a resubmission of an earlier submission. The following is a list of the peer review reports and author responses from that submission.
Round 1
Reviewer 1 Report
The authors added additional information in agreement with their conclusion. furthermore they clarified the mistaken concerning the picture S1, including the correct image and the dataset as supplementary materials. The revised manuscript fulfilled all my worries and can be considered for the publication in Pharmaceuticals Journal.
Author Response
We would like to express our sincere gratitude to Reviewer 1 for the constructive criticism and patience with our previous mistake. Apart from our responses in the last round, several other minor changes have also been made as follows:
Line 68: “endpoints” reworded as “markers” for consistency across the manuscript.
Line 76: full stop added.
Table 1 title: abbreviation "(SCQ)” inserted.
Lines 44-45: brackets moved upwards for clarity of explanation.
Line 68: figure number corrected as “2e”.
Line 89: figure number corrected as “2f”.
Line 90: section title reworded as “Redox Activity” for consistency across the manuscript.
Line 182-188: texts (originally from 2.5.2) moved upwards.
Line 201-229: section “Oxidative Protein and DNA damage" (originally 2.5.2) is split as “Oxidative Protein Damage” and “Oxidative DNA damage” (now 2.5.2 and 2.5.3). Now each subsection/paragraph is reflected by two respective sub-figures from Figure 2 to 5 and allows reading with ease.
Reviewer 2 Report
The presented manuscript is a follow-up study of cytoprotective activity of short chain quinones (SCQ) that is maintained through prevention of mitochondrial disfunction. The previous work is well cited, in this new manuscript we have extended number of compounds and more detailed investigation of the mechanism of action. In my opinion, the majority of findings here is new.
The autors did a number of experiments that revealed their mechanism of action. They evaluated the molecules ability of Cytoprotection against Mitochondrial Dysfunction using cells with rotenon reduced viability and found 62 compounds with improvement out of that nine compounds had a significant impact. Some of the compounds restore the level of ATP. The authors measured the reduction of the amount of extracellular lactate and hydroxybutyrate but did not find a correlation between them and the cytoprotection. Similarly, they did not find a correlation between basal lipid peroxidation, or Lin28A expression, or cellular Hsp70 expression, or tubulin acetylation levels, or oxidative protein- or DNA-damage and cytoprotection.
It seems, that most of the evaluated properties did not correlate with the cytoprotection except from the enzymatic reduction of SCQs and the acute redox-dependent rescue of ATP and there are likely some other mechanisms by which the compounds perform.
The authors discuss this in detail within their conclusion part and come with some hypotheses about the activity.
The article is metodically well done, experiments support the final conclusions although some of them disproved the original hypotheses. This is highly valuable that the authors publish both positive and negative results. The article is easy to follow and I believe that it will be very interesting for the readers of the journal.
I have reviewed this article for another journal that did not accept it for publishing and since the authors have included all my suggestions, I do not have any additional requirements or negative comments.
Author Response
We would like to express our sincere gratitude to Reviewer 2 for the thoughtful comments. Apart from our previous responses in the last round, several other minor changes have also been made as follows:
Line 68: “endpoints” reworded as “markers” for consistency across the manuscript.
Line 76: full stop added.
Table 1 title: abbreviation "(SCQ)” inserted.
Lines 44-45: brackets moved upwards for clarity of explanation.
Line 68: figure number corrected as “2e”.
Line 89: figure number corrected as “2f”.
Line 90: section title reworded as “Redox Activity” for consistency across the manuscript.
Line 182-188: texts (originally from 2.5.2) moved upwards.
Line 201-229: section “Oxidative Protein and DNA damage" (originally 2.5.2) is split as “Oxidative Protein Damage” and “Oxidative DNA damage” (now 2.5.2 and 2.5.3). Now each subsection/paragraph is reflected by two respective sub-figures from Figure 2 to 5 and allows reading with ease.
Reviewer 3 Report
This article “Bioactivity Profiles of Cytoprotective Short-Chain Quinones”. The comments for this article are as follows:
- Owing to there are many compounds in this manuscript (Table 1) that have been published previously, the authors should not write 103 novel short-chain quinones in the "abstract" section. Although the authors marked the reference source in the manuscript (Table 1), it will confuse the readers, as this manuscript contains 103 novel short-chain quinones. In fact, there should be only 87 of them that have not been published, and it is strongly recommended to modify it. Otherwise, there is a suspicion of submitting more than one manuscript.
- This manuscript is similar to the paper of previous author's published in Pharmaceuticals 2020, 13, 29; doi:10.3390 and Pharmaceuticals 2020, 13, 184, pharmaceuticals-13-00184-v2. For example, the measurement of ATP has been tested before, and only the concentration was changed in this time. Do the authors find any valuable results? If it does, please explain it clearly in "discussion" section.
- Figure 5 is recommended to be removed. In this figure, reviewer could not see any valuable information inside. I think that is not meaningful to readers.
- Basically, this manuscript let the reviewer thinks that this is a research report which has no high quality. In this manuscript, I can't feel the excellent research methods and important findings, or let the readers have more knowledge. The experimental results neither found any specific mechanism of action, nor can it explain that those quinone compounds are worth developing. This manuscript is like a pieced together monster, I don't know how it will benefit the readers, and it is not worth publishing.
- I suggest to do a few more obvious and meaningful experiments, without using the 103 compounds to confuse the readers. Major discoveries can be found when the key mechanisms are made, why do you not to make it?
- In addition, the article format is poor. Please standardize the format of references. Do not together use in uppercase and lowercase in every "word" for the title of the references. For example, reference 2, 3, 7, 8, 12, 16, 17, 18, 22, 24 , 30, 45, 64, 68, and 69. The format should be the same with the other references.
Author Response
We would like to express our sincere gratitude to Reviewer 3 for the constructive criticism. Revisions have been made according to the comments or suggestions as follows.
Point 1: Owing to there are many compounds in this manuscript (Table 1) that have been published previously, the authors should not write 103 novel short-chain quinones in the "abstract" section. Although the authors marked the reference source in the manuscript (Table 1), it will confuse the readers, as this manuscript contains 103 novel short-chain quinones. In fact, there should be only 87 of them that have not been published, and it is strongly recommended to modify it. Otherwise, there is a suspicion of submitting more than one manuscript.
Response 1: We appreciate the reviewer’s comment and have removed or reworded all “novel” wordings from the abstract, main texts and supplementary materials. Of the 103 SCQs included in the current manuscript, 78 were assessed in a single cytoprotection assay against rotenone (Woolley et al 2019), 16 were assessed for metabolic stability (Feng et al 2020a) and 12 were assessed for cytotoxicity without rotenone presence (Feng et al 2020b). The references have been clearly incorporated into Table 1 which allows reading with ease. For better clarification, above numbers have been stated in Lines 59-62. Here, we declare that all data included in the current manuscript are new and have never been incorporated to any other manuscripts or published materials.
Point 2: This manuscript is similar to the paper of previous author's published in Pharmaceuticals 2020, 13, 29; doi:10.3390 and Pharmaceuticals 2020, 13, 184, pharmaceuticals-13-00184-v2. For example, the measurement of ATP has been tested before, and only the concentration was changed in this time. Do the authors find any valuable results? If it does, please explain it clearly in "discussion" section.
Response 2: We appreciate the reviewer’s comment but we respectfully disagree that measurement of acute rescue of ATP levels was tested before. The aim of our previously published paper (Feng et al. 2020b) is very different from the current manuscript. The former paper established a detailed toxicology profile for selected cytoprotective SCQs including their inherent metabolic toxicity (i.e. ATP, NAD(P)H, etc) in the ABSENCE of rotenone. In contrast, the current manuscript aimed to illustrate whether acute rescue of ATP levels confers cytoprotection against mitochondrial dysfunction (which was done in the PRESENCE of rotenone). In addition, the two cell culture systems of the respective paper and manuscript are different (Feng et al. 2020b; Woolley et al. 2019). For all parameters assessed in the current manuscript assessed, all 103 SCQs are deliberately assessed at 10 µM, which replicates what was used in the original cytoprotection assay that only used a small number of compounds (Woolley et al 2019), to identify any parameters associated with cytoprotection. In the current manuscript, our results revealed the mild correlation of acute rescue of ATP levels and SCQ-induced cytoprotection, whereas importantly this does not necessarily imply that the acute rescue of ATP levels is cytoprotection per se. In fact, this is merely a reflection of the ability of SCQs to interact with NQO1 and to be reduced. This discussion has been stated in Lines 275-284 and 361-368. To better support the flow in the Discussion section, Figure 7 has been moved upwards right after the paragraph where it was first referred to (Lines 282 and 297-300).
Point 3: Figure 5 is recommended to be removed. In this figure, reviewer could not see any valuable information inside. I think that is not meaningful to readers.
Response 3: We appreciate the reviewer’s comment but we respectfully disagree that this section 2.5 or Figure 5 is devoid of valuable information. Similar to other parameters that endpoints assessed in the current manuscript (i.e. extracellular lactate, β-hydroxybutyrate, expression of Lin28A or Hsp70, HDAC6 inhibition), effects on oxidative damage does not positively correlate with SCQ-induced cytoprotection. Even though, all these results, including section 2.5, suggest that other mechanisms are responsible for the cytoprotective effects of SCQs. In this particular case, we provide evidence that cytoprotection by SCQs is not driven by antioxidant function as previously thought. Importantly, results in section 2.5 highlighted the superior safety features of the majority of our SCQs.
Point 4: Basically, this manuscript let the reviewer thinks that this is a research report which has no high quality. In this manuscript, I can't feel the excellent research methods and important findings, or let the readers have more knowledge. The experimental results neither found any specific mechanism of action, nor can it explain that those quinone compounds are worth developing. This manuscript is like a pieced together monster, I don't know how it will benefit the readers, and it is not worth publishing.
Response 4: We appreciate the reviewer’s comment but respectfully disagree that the findings are not important. Instead, we propose an unexpected mechanism for SCQs that induces cytoprotection via NQO1-dependent signaling, which could completely change the way we see redox active compounds: not as antioxidants but as signaling molecules. For several SCQs with high cytoprotection, stability and safety, our in vivo data is under submission, in preparation or accepted for publication. We also disagree that these results are not worth publishing. In fact, we are in contact with several Pharmaceutical companies for the commercialization of selected SCQs.
For the manuscript structure, significant changes have been made accordingly (Lines 76-298). In the Results section, main texts and plot figures had already been separated and characterized into 5 parts as follows: 1) cytoprotection against mitochondrial dysfunction, 2) effects on metabolism-related markers, 3) chemical reduction of test compounds, 4) expression of cytoprotective proteins, and 5) effects on oxidative damage. For even better manuscript structure, the original section 2.5 “oxidative protein and DNA damage” has been split as “oxidative protein damage” and “oxidative DNA damage”. Similarly, original texts have also been split into the two new sub-sections 2.5.2 and 2.5.3. As such, each subsection/paragraph is reflected by two respective sub-figures from Figure 2 to 5 (Lines 24-229). This clear and logical flow in the Results section allows reading with ease.
Point 5: I suggest to do a few more obvious and meaningful experiments, without using the 103 compounds to confuse the readers. Major discoveries can be found when the key mechanisms are made, why do you not to make it?
Point 5: We appreciate the reviewer’s comment but respectfully disagree that the experiments included in the current manuscript were unobvious or meaningless, or that our 103 compounds would confuse the readers. Please refer to the explanations detailed in Responses 1-4. Considering the significant amount of results generated for the current manuscript (also suggested by reviewers in prior submissions), we plan to further address the detailed mode of action of this NQO1-dependent signaling in our next experiments with a single reference compound. Based on the expected number of experiments to be performed, this approach would be significantly beyond the scope of the current manuscript as it will employ pull down assays (with antibodies and newly synthesized biotin-SCQ complexes), mass spectrometry, western blotting and the use of RNAi to identify how the presence of SCQs will affect NQO1 complex composition. This possibly lengthy approach has the potential to illuminate how NQO1 can affect cell viability in the presence of rotenone. The discussion has been revised to address this concern of the reviewer (Lines 378-384).
Point 6: In addition, the article format is poor. Please standardize the format of references. Do not together use in uppercase and lowercase in every "word" for the title of the references. For example, reference 2, 3, 7, 8, 12, 16, 17, 18, 22, 24 , 30, 45, 64, 68, and 69. The format should be the same with the other references.
Response 6: We appreciate the reviewer’s comment and agree with the reviewer. The referencing style has been updated with MDPI Referencing Style according to the Guidelines for Authors from Pharmaceuticals, and the irregularities of uppercase and lowercase have been manually corrected.

Round 2
Reviewer 3 Report
I must to say that synthesizing 103 different kinds of short-chain quinones is really a big process. But this has nothing to do with the results of my review. Although the authors disagree with the reviewer's opinions, and made many explanations. Here I will give the authors another piece of advice. I believe that the authors have spent a lot of money and time on this manuscript, but from my many years of experimental experience, I still think that 103 kinds of short-chain quinones are actually the denominator (make the experiment seem to have done a lot), but it is the numerators that are really useful, but there are no more than 25. Therefore, most of the experimental results that are not valid which can be put in the supplementary information at most. Of course, every journal has their own opinions, and I all respect them. To be honest, from Figure 2 to Figure 6 the performance of these different compounds in these tested substances, what can readers actually get? In my opinion, the amount of screening is large, but there are few practical ones. Figure 1 is a sufficient figure (although the synthesis project is really huge). Because the authors do not explain which compounds are useful? Why is it effective? Even if the authors disputes Figure 7. I want the authors to know that a picture of the signal transduction pathways is not simple. You can go to JBC (Journal of Biological Chemistry) to view the map of the other reasearcher's results about signal transduction pathways, which have been verified by many experiments. By this standard, Figure 7 is definitely a failure and vague. You have to know that it is not only these substances you determin that are acting in the cell. In fact, the reactions in the entire cell are quite complicated and cannot be as simple as the one shown in Figure 7. At the same time, the authors also mentioned that some specific compounds (Short-chain quinones) have been negotiated with pharmaceutical companies to commercialize them. To be honest, based on the experimental level of this manuscript, I believe that pharmaceutical companies should not be interested. So the authors mentioned that some in vivo data is being prepared for submission. This means that those useful compounds are the protagonists. I believe that drug manufacturers should be more interested in those compounds. Then I would like to ask the authors why you don’t prepare it all at once, and submit an experimental results with completed, accurated, corrected, and sufficient evidence. This should be able to be submitted to a higher-level journal? Why do you have to segment it like this? What is the contribution? Is it for the number of papers? I have checked the papers in your laboratory, and some papers published in "Scientific Reports". The standards of molecular biology are also good. This means that you have the ability to do better. Why not maintain the standards? I don't understand the reasons. I have seen that the author’s revised version is indeed better, but my opinion has been stated. I still maintain the original judgment.